

# The role of T cells in the treatment of Parkinson's disease

Zhuomiao Lin[1,*], Xihui Yu[2,*], Yunming Zhong[1], Guozhu Tan[3] and Jiahong Zhong[1]

[1] Department of Clinical Pharmacy, Meizhou People's Hospital (Huangtang Hospital), Meizhou, Guangdong, China
[2] Department of Pharmacy, The Second Affiliated Hospital of Shantou University Medical College, Shantou, Guangdong, China
[3] Department of Orthopaedics and Traumatology, The Seventh Affiliated Hospital, Southern Medical University, Foshan, Guangdong, China
[*] These authors contributed equally to this work.

Corresponding author
Jiahong Zhong, zhongjiahong@mzrmyy.com

## ABSTRACT

Current pharmacological treatment of Parkinson's disease (PD) predominantly employs dopaminergic agents aimed at enhancing cerebral dopamine levels. While these therapeutic strategies provide symptomatic relief, their palliative nature is frequently associated with dose-dependent complications, including gastrointestinal disturbances, emetic symptoms, and motor complications such as dyskinesia. Moreover, the honeymoon period of drugs has greatly limited their clinical application. The multifactorial etiology of PD continues to challenge researchers, yet substantial evidence implicates α-synuclein as a critical pathogenic mediator. Emerging findings suggest that dysregulated neuroimmune interactions constitute a fundamental mechanism in PD progression, where chronic immune activation appears particularly detrimental to neuronal survival. Notably, neuroinflammatory cascades coupled with compromised blood-brain barrier (BBB) integrity create a self-perpetuating cycle of neural degeneration, wherein α-synuclein-specific T cells exacerbate disease pathology while regulatory T cell populations demonstrate potential immunomodulatory capacities. This review systematically examines the mechanistic interplay involving neuroinflammatory cascades, BBB compromise, central nervous system (CNS) immunoregulation, and T lymphocyte subpopulations (including regulatory T cells) in the pathogenesis of PD. By synthesizing current evidence, we aim to establish a conceptual framework supporting the investigation of cellular immunity-based therapies for PD.

## INTRODUCTION

Parkinson's disease (PD) is recognized as one of the four leading neurodegenerative disorders worldwide. In China's population aged ≥70 years, epidemiological studies reveal an incidence range of 3–5%, with case numbers demonstrating an age-dependent escalation pattern (*Ben-Shlomo et al., 2024*; *Tanner & Ostrem, 2024*; *Su et al., 2025*). The neuropathological profile of PD is defined by early degeneration of substantia nigra pars compacta (SNpc) dopaminergic neurons, with subsequent dopamine depletion

in basal ganglia circuitry driving core motor deficits such as bradykinesia, resting tremor, and postural instability. Non-motor manifestations including hyposmia and rapid eye movement (REM) sleep behavior disorder often emerge as prodromal indicators prior to motor symptom onset. Pathognomonic Lewy bodies, characterized by aggregated α-synuclein (α-Syn) protein deposits, are predominantly localized to pigmented nuclei—notably the substantia nigra and locus coeruleus (*Morris et al., 2024*). PD management primarily relies on pharmacotherapies targeting dopaminergic enhancement through dopamine level elevation or receptor stimulation. However, these interventions demonstrate limited neuroprotective efficacy against dopaminergic neuron degeneration and are frequently associated with treatment-related complications including the "wearing-off phenomenon" and "dyskinesia" (*Marsili et al., 2025*; *Yan et al., 2025*).

The multifactorial etiology of PD encompasses genetic predisposition, aging processes, oxidative stress mechanisms, and environmental influences, though the precise pathophysiological cascade remains incompletely elucidated. Emerging evidence implicates sustained activation of neuroinflammatory pathways as a pivotal mechanism in PD progression, with chronic immune system overactivation potentially serving as a proactive contributor to neuronal apoptosis rather than merely representing a secondary consequence (*Ben-Shlomo et al., 2024*). Neuroinflammation constitutes a pivotal mechanism underlying the pathogenesis of multiple neurodegenerative disorders, including multiple sclerosis (MS), Alzheimer's disease (AD), and PD (*Voet et al., 2019*). In PD patients, intense inflammatory activation with elevated cytokine concentrations is predominantly observed in the substantia nigra of the midbrain. Blood–brain barrier (BBB) breakdown synergizes with neuroinflammatory cascades to exacerbate pathological alterations in brain parenchyma. Peripheral immune dysregulation-marked by lymphocyte subset imbalance and BBB disruption-enables T cell infiltration into affected brain regions in PD, thereby amplifying neuronal vulnerability. α-Syn has been demonstrated to play a pivotal and crucial role in the pathogenesis of PD (*Brochard et al., 2009*; *Huang et al., 2022*; *Xu et al., 2023*; *Ma et al., 2024*). It serves as the key factor that activates and promotes inflammation and neurodegeneration in human PD. α-Syn-specific T cells exacerbate the pathological changes, indicating that cellular immunity is of great significance in the progression of PD (*Lindestam Arlehamn et al., 2020*). As pivotal immunomodulatory elements, regulatory T cells (Tregs) critically sustain immune tolerance through dynamic suppression of T lymphocyte activation and proliferation. This immunosuppressive capacity enables Tregs to orchestrate immune homeostasis, positioning them as a critical investigative focus for PD therapeutic development. Growing evidence underscores the critical involvement of cellular immunity in PD pathogenesis, particularly given the current absence of disease-modifying pharmacotherapies. The present work systematically examines the pathophysiological interplay involving neuroinflammatory cascades, BBB integrity loss, CNS immune regulation, and T cell subpopulations (including Tregs) during PD progression, establishing an evidence-based framework for advancing immunotherapeutic strategies. These mechanistic insights highlight the transformative potential of cellular immunity in developing novel therapeutic paradigms for PD management.

## SURVEY METHODOLOGY

This investigation employed a systematic document retrieval across PubMed (https://pubmed.ncbi.nlm.nih.gov/) and Google Scholar databases (https://scholar.google.com/), followed by integrative scientometric analysis of T cell involvement in Parkinson's disease. This study focuses on the role of T cells in the pathogenesis and development of PD, and explores the possibility of cellular immunity as a therapeutic method for PD. The keywords included "Parkinson's disease", "T cells", "cellular immunity", "inflammation", "regulatory T cells" and "α-Syn", performed by combining these descriptors using the Boolean operators "OR" and "AND". Meets the inclusion criteria were restricted to peer-reviewed literature (original research articles and scientific briefs) published in English from 1988 to 2025. Exclusion criteria comprised non-peer-reviewed materials (monographs, dissertations, book chapters, conference proceedings without peer review) and secondary literature including review articles.

### PD and neuroinflammation

The immune system mobilizes inflammatory processes as its frontline defense against acute pathogen exposure, functioning to preserve tissue integrity through microbial exclusion and wound repair facilitation. Paradoxically, when such responses transition into chronic activation states, they instigate maladaptive immunopathological cascades culminating in host tissue injury. Within the CNS compartment, microglia fulfill dual immunoregulatory roles: executing innate phagocytic clearance and antigen presentation, essential for neuroprotective surveillance. While the BBB historically conferred CNS immune privilege, emerging neuroimmunological paradigms now implicate dysregulated neuroinflammation as a pivotal etiological contributor to progressive neurodegenerative disorders, notably MS, AD, and PD (*Kim & Joh, 2006*). Neuropathological investigations of midbrain and substantia nigra in PD cases demonstrate intense focal neuroinflammation mediated by reactive microglia and astrocytes. Positron emission computed tomography (PET) imaging reveals microglial activation patterns with region-specific neuroinflammatory amplification, particularly in basal ganglia and striatal structures, revealing temporal decoupling relative to clinical disease staging (*Ouchi et al., 2005*; *Gerhard et al., 2006*). Accumulating evidence from postmortem examinations, neuroimaging modalities, and biofluid biomarker analyses consistently identifies neuroinflammation as a defining pathophysiological feature in PD (*Imamura et al., 2003*). PD pathophysiology involves microglial activation manifested through population expansion, morphological transformation, and phenotypic alteration. Concurrent biochemical analyses of CNS compartments reveal elevated cytokine concentrations in both brain parenchyma and cerebrospinal fluid (CSF), indicative of sustained neuroinflammatory activity underlying PD progression (*King & Thomas, 2017*). Neuroinflammatory activation drives neurodegeneration in nigrostriatal pathways during PD progression. Nevertheless, the immunological origins of early-stage proinflammatory signaling remain ambiguous, potentially arising from either deficient immunoregulation (functional inactivation) or pathological acquisition of inflammatory phenotypes (functional enhancement).

## PD and microglia

Contemporary investigations employing longitudinal PD modeling and systematic analyses of postmortem specimens have revealed that microglial maladaptation precedes neuronal demise in PD-affected brain regions, emerging even prior to detectable neuronal pathology (*Sanchez-Guajardo et al., 2013*). Notably, microglial activation with concomitant proliferative responses persists in PD-affected brain regions devoid of significant neuronal degeneration postmortem (*Imamura et al., 2003*). This neuroinflammatory phenomenon has been quantitatively validated through PK11195-targeted PET imaging modalities, demonstrating preserved microglial reactivity independent of neurodegenerative progression (*Ouchi et al., 2005*).

Individuals with rapid eye movement sleep behavior disorder (RBD), recognized as a significant risk factor for PD, exhibit microglial hyperplasia and activation several years prior to clinical PD diagnosis. These microglial changes precede clinical PD diagnosis and play a role in disease progression, suggesting their involvement in early pathogenic mechanisms (*Stokholm et al., 2017*). Epidemiological evidence indicates that chronic ibuprofen use, a non-steroidal anti-inflammatory drug, confers a 35% reduction in PD risk (*Chen et al., 2005*). Early-stage PD patients demonstrate region-specific microglial activation predominantly localized to the midbrain substantia nigra, with activation intensity positively correlating with clinical severity scores and dopaminergic terminal degeneration. This neuroinflammatory signature remains independent of disease duration, suggesting its potential as a stage-specific biomarker (*Orr et al., 2005*), supporting the hypothesis that neuroinflammatory processes drive dopaminergic neuron degeneration.

Microglial activation represents a critical component of PD-associated inflammatory and immune responses. Under normal physiological conditions, dopaminergic neurons hardly express major histocompatibility complex I (MHC-I). However, under pathological or specific stimuli, its expression is significantly upregulated. The upregulation of MHC-I enables dopaminergic neurons to present antigens (such as misfolded α-Syn) to CD8+ T cells (*Harms, Ferreira & Romero-Ramos, 2021*; *Wang et al., 2021*). In the PD model, MHC-I positive neurons are attacked by T cells, resulting in neuronal death. In PD models with overexpression of MPTP or α-Syn, the upregulation of MHC-I is positively correlated with neuronal death. Inhibiting interferon-γ (IFN-γ) signaling or enhancing the degradation pathway of MHC-I can reduce neuronal injury (*Sulzer et al., 2017*; *Garretti et al., 2019*). Whereas neurons lack major histocompatibility complex II (MHC II) expression, microglia function as MHC class II antigen-presenting cells in the CNS. MHC II molecules Microglia and similar immune cells specialize in taking in aggregated proteins (like α-Syn) and presenting their fragments *via* MHC II. These properties position microglia as key antigen-presenting cells for MHC class II-restricted α-Syn peptides to T lymphocytes. MHC II expression on microglial surfaces functions as a molecular signature for antigen presentation, positioning these cells as critical mediators of T-cell activation (*Scheu et al., 2019*; *Tschoe et al., 2020*; *Wang et al., 2024*). Genome-wide association studies (GWAS) implicate PD-risk loci with enriched expression in immune cells and functional annotations linked to immune regulatory pathways. A notable example is leucine-rich repeat kinase 2 (LRRK2), which modulates autophagic processes in immune cells (*Nalls et al., 2014*).

Clinical and preclinical evidence highlights a bidirectional relationship between genetic susceptibility, protein aggregation, and neuroinflammation in PD. Specifically, α-Syn aggregation triggers both innate and adaptive immune responses (*Ransohoff, 2016*), while neuroinflammatory processes reciprocally accelerate α-Syn misfolding. This reciprocal interaction is supported by convergent data from GWAS and mechanistic studies (*Gao et al., 2008*). These interdependent processes form a self-perpetuating pathological cycle in PD. Chronic inflammatory signaling serves as a primary driver of disease progression, promoting dopaminergic neurodegeneration. Neuroinflammation is characterized by T-cell infiltration and microglial activation, which generate cytotoxic immune responses that further exacerbate neuronal damage (*Von Euler Chelpin & Vorup-Jensen, 2017*).

## PD and T cells

Activated glia secrete pro-inflammatory cytokines including interleukin-1β (IL-1β) and tumor necrosis factor-α (TNF-α) (Fig. 1), enhanced expression of adhesion molecules in microvascular endothelial cells has been demonstrated to promote elevated BBB permeability. This facilitates the infiltration of peripherally activated helper T cells (Th1/Th2 subsets) into brain inflammatory sites, exacerbating neuronal injury and accelerating disease progression. The initial identification of CD3+ T-cell infiltration in the CNS of PD brains was reported by *McGeer et al. (1988)*, using immunohistochemical detection of this pan-T-cell marker. Brochard et al. further demonstrated that CD4+ and CD8+ T cells specifically infiltrate the midbrain substantia nigra in PD patients, excluding B lymphocytes and natural killer (NK) cells. Immunohistochemical analysis revealed these T cells localized adjacent to blood vessels and neuromelanin-containing dopaminergic neurons, displaying characteristic lymphocyte morphology confirmed by electron microscopy. Quantitative analysis revealed a 10-fold elevation in CD8+ and CD4+ T-cell densities within the SNpc of PD cases relative to age-matched controls, whereas no significant changes were detected in the unaffected red nucleus (*Brochard et al., 2009*).

The selective accumulation of T cells in diseased brain areas supports the hypothesis of active transendothelial migration rather than non-specific effects of BBB permeability induced by systemic inflammation. Elevated T-cell percentages in CSF and peripheral blood of PD patients suggest modifications in T-cell subsets. This peripheral T cell compartment remodeling facilitates their recruitment across the BBB and subsequent accumulation within the midbrain substantia nigra in PD (*Brochard et al., 2009*; *Sommer et al., 2018*). Quantitative analysis demonstrated marked reduction of neuromelanin-positive neurons in the PD substantia nigra, co-occurring with perineuronal T-cell infiltration that was not observed in control groups. The T lymphocyte-to-neuromelanin-containing neuron ratio in PD patients was approximately 1:2, compared to 1:10 in healthy controls (*Sulzer et al., 2017*; *Sommer et al., 2019*). Furthermore, autopsy studies have shown that the number of melanin neurons in the substantia nigra of PD patients is reduced by approximately 50–70% compared to healthy individuals, and this reduction is associated with disease progression (*Rudow et al., 2008*). Flow cytometry characterization of peripheral blood mononuclear cells identified a significant increase in IL-17+ CD4+ T-cell frequency among PD patients (*Sommer et al., 2019*). Postmortem neuropathological evaluations further

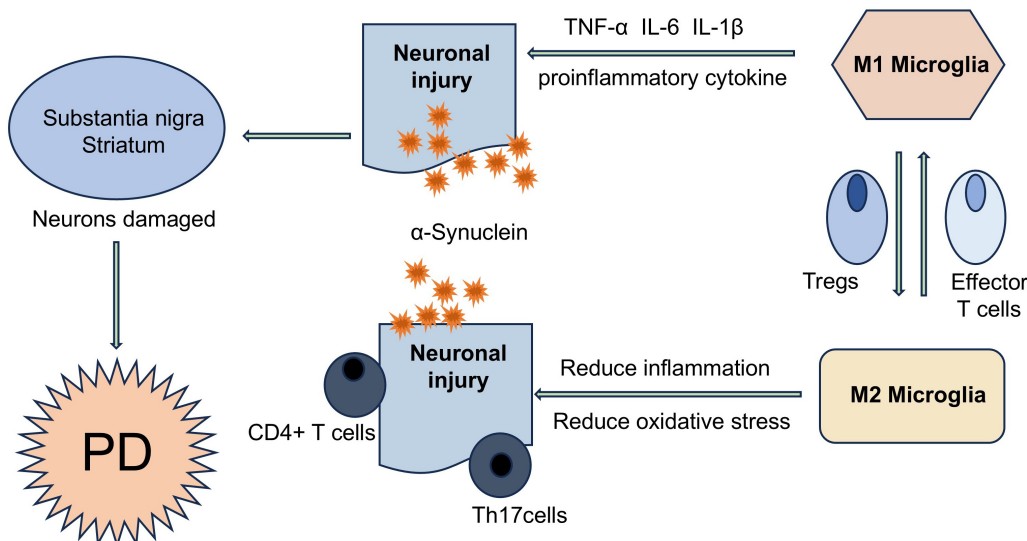

**Figure 1** **T cells are involved in the disease process of PD in different ways.** α-Syn aggregates can present and activate T cells *in vivo*, further triggering autoimmune inflammation, resulting in the imbalance of effector Th cell subsets (such as Treg and Th17 cells), and ultimately accelerating the PD disease process. T cells and neurons can directly interact with each other through LFA1- ICAM signaling or Fas-FasL signaling.

highlighted adaptive immune system engagement, marked by augmented T-lymphocyte infiltration in brain parenchyma and elevated systemic Th17 cell proportions (*Benner et al., 2008*; *Sommer et al., 2019*). Mechanistic investigations utilizing induced pluripotent stem cell (iPSC)-derived midbrain dopaminergic neurons revealed that co-culture with activated T cells or recombinant IL-17 exposure promoted neuronal apoptosis through IL-17 receptor (IL-17R) upregulation and nuclear factor kappa-B (NF-κB) signaling activation. Importantly, therapeutic interventions targeting the IL-17 axis-including cytokine neutralization, receptor blockade, or monoclonal antibody therapy-attenuated neuronal degeneration, thereby establishing a causal role for IL-17-secreting T lymphocytes in sporadic PD pathogenesis (*Sommer et al., 2018*) (Fig. 2).

Circulating self-reactive T cells in PD patients, which breach central tolerance mechanisms, gain access to the CNS through a compromised BBB, as demonstrated by murine models of T cell infiltration. 1-Methyl-4-phenyl-1,2,3,6-tetrahydropyridine (MPTP)-treated murine models demonstrated T-cell activation and infiltration within midbrain substantia nigra and striatal regions, mirroring pathological hallmarks observed in human PD (*Chandra et al., 2017*; *Liu et al., 2017*). Independent investigations using microglial proliferation murine models confirmed T-cell accumulation in the midbrain substantia nigra as early as 48 h post-MPTP administration (*Depboylu et al., 2012*). Immunodeficient murine strains lacking functional T lymphocytes (Rag1$^{-/-}$ and Tcrb$^{-/-}$ mice) displayed resistance to MPTP-induced dopaminergic neurodegeneration, thereby establishing a causal link between T-cell dysfunction and nigrostriatal damage. Cd4$^{-/-}$ mice, deficient in CD4+ T lymphocytes, also demonstrated protection against dopaminergic

**Peripheral system**

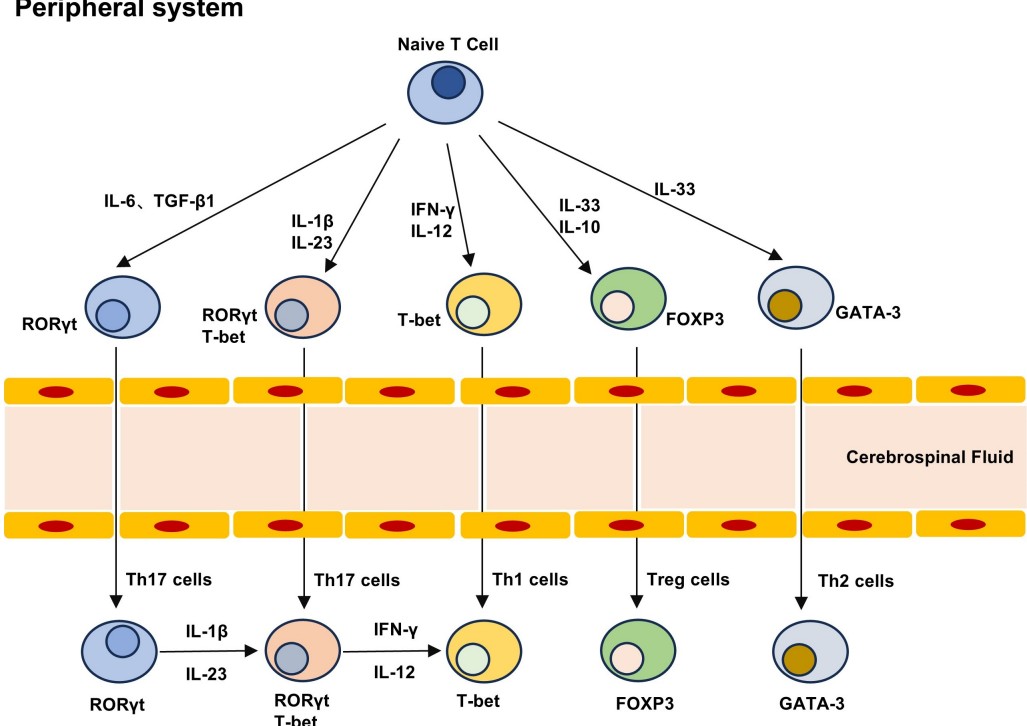

**Intracerebral parenchyma**

**Figure 2  T cell differentiation lineage between peripheral system and brain parenchyma.** Naive T cells can differentiate into different subpopulations under different cytokines, including pro-inflammatory Th1 and Th17 cells and anti-inflammatory Th2 and Treg cells. The development and maturation of T cell subsets require specific transcription factors, and Th17 cells can produce highly pathogenic Th17 cells and low pathogenic Th17 cells that do not express T-bet under the stimulation of IL-23 and IL-1β. Treg cells in the brain can not only play an immune function, but also play a role in repair neurogenesis and angiogenesis.

neuron loss, highlighting the critical contribution of this adaptive immune subset. Additionally, SCID mice (deficient in both T and B lymphocytes) displayed reduced neuronal death following MPTP administration, whereas reconstitution with wild-type splenocytes restored MPTP-induced neurodegeneration. Experimental data reinforce the pivotal role of T cell-driven immunity in the pathogenic cascade of PD (*Benner et al., 2008*; *Brochard et al., 2009*).

## PD and cytokines

Recent investigations have uncovered the critical role of chemokines in sustaining nigrostriatal pathology. Although pro-inflammatory molecules such as C-C motif chemokine ligand 5 (CCL5) and CCL11 are involved, inflammation alone fails to drive persistent pathological processes in the nigrostriatal system. Administration of tumor necrosis factor-α (TNF-α) and interleukin-1β (IL-1β)-two well-characterized pro-inflammatory cytokines-failed to elicit chronic neurodegeneration in MPTP-treated

murine models (*Chandra et al., 2017*). Overexpression of human α-Syn *via* Adeno-associated virus (AAV) vectors in murine substantia nigra neurons induced microglial activation, production of inflammatory cytokines, and adaptive immune engagement. These pathological events emerged as early as 2 weeks post-injection, coinciding with minimal to absent dopaminergic neuronal loss, and were spatially restricted to α-Syn-overexpressing regions. Collectively, these findings establish α-Syn as a direct activator of microglia and adaptive immunity *in vivo*, independent of cell death mechanisms. In rAAV2/7-α-Syn-injected rats treated with FK506-a calcineurin inhibitor that suppresses T cell signaling and interleukin-2 (IL-2) transcription-immunomodulation reduced nigral microglia, macrophages, and CD4+ T cell infiltration while delaying CD8+ T cell accumulation. This treatment strategy dose-dependently enhanced dopaminergic neuron survival and demonstrated a positive trend in motor function improvement (*Theodore et al., 2008*; *Vander Perren et al., 2015*). The essential role of an intact immune system-specifically CD4+ T cells, in mediating neurodegeneration within MPTP-induced Parkinson's disease models is underscored by these findings.

Genetic evidence linking PD-causing variants to immune signaling pathways strengthens the evidence for T-cell involvement in Parkinson's disease pathogenesis. Notably, PTEN induced putative kinase 1 (PINK1) and Parkin-autosomal recessive PD-associated genes-carry loss-of-function variants that predispose to autosomal recessive PD, particularly early-onset cases. These genes are postulated to modulate the suppression of mitochondrial antigen processing (MitAP). Functional inactivation of PINK1/Parkin results in accumulation of mitochondrial antigens displayed by major histocompatibility complex I (MHC I) molecules on dendritic cells (DCs) and macrophages. Elevated MitAP expression in antigen-presenting cells (APCs) promotes the differentiation of antigen-specific T lymphocytes. During aging, neuroinflammatory processes and microglial activation enhance T-cell migration into Parkinson's disease-affected brain regions, where cytotoxic (CD8+) T cells mediate dopaminergic neuron damage *via* MHC I-restricted antigen presentation. LRRK2, a gene mutated in late-onset PD, encodes a kinase expressed in Th17 cells and cytotoxic T lymphocytes, with upregulation observed during immune challenges. Aberrant LRRK2 activity is associated with alterations in bone marrow myelopoiesis, peripheral blood myeloid cell differentiation, and intestinal immune homeostasis. LRRK2 mutations potentiate a pro-inflammatory phenotype in peripheral immune cells, characterized by enhanced cytokine production and chemotactic responsiveness (*Wallings & Tansey, 2019*; *Sharma et al., 2025b*). This neuroinflammatory milieu exerts cytotoxic effects on substantia nigra dopaminergic neurons, driving neurodegenerative progression (*Park et al., 2017*).

## Breakdown of the BBB in PD

The CNS has long been recognized as an immunologically privileged site, a phenomenon largely attributed to the BBB-a dynamic interface constituted by endothelial cells, pericytes, and astrocytes. This specialized barrier system comprises two functional components: the endothelium-mediated barrier between brain capillaries and parenchyma, and the choroid plexus epithelium separating plasma from CSF. Collectively, these structures

restrict the passage of blood-derived proteins, antibodies, immune cells, and xenobiotics into brain tissue. In addition to regulating brain homeostasis and shielding neural circuits from circulating pathogens, the BBB serves a pivotal function in upholding CNS integrity, a function essential for maintaining optimal neurological function (*Daneman & Prat, 2015*). In chronic inflammation, disruption of endothelial tight junctions—the primary structural determinants of paracellular permeability-couples with elevated peripheral cytokine concentrations targeting BBB-forming CNS endothelial cells. This dual insult induces endothelial permeability, facilitating immune effector cell extravasation into the CNS parenchyma (*Varatharaj & Galea, 2017*). Inflammatory signaling triggers CNS endothelial cells to selectively express leukocyte-trafficking receptors (ICAM-1, VCAM-1), creating molecular scaffolds for the sequential capture of circulating leukocytes (T cells, monocytes) and vascular permeability mediators (caveolin-1, PLVAP). This process facilitates transcellular trafficking of immune effectors across the BBB, rendering the CNS parenchyma vulnerable to infiltrating leukocytes, autoantibodies, and pro-inflammatory cytokines. As observed in multiple sclerosis models, T cells exploit BBB disruption to transmigrate into the CNS. Perivascular microglia maintain constitutive bidirectional crosstalk with endothelial cells, continuously surveying BBB integrity and blood-derived molecular influx (*Chen et al., 2019*; *Haruwaka et al., 2019*; *Huang et al., 2020*). Studies demonstrate spatiotemporally coordinated activation of vascular endothelium, BBB breakdown, and microglial activation in neurological disorders. During CNS disease progression, microglia dynamically respond to BBB perturbations by altering morphology, functional phenotypes, and activation states, thereby exerting reciprocal regulatory effects on BBB permeability. This bidirectional interaction is a dynamic process involving multiple molecular mediators and signaling pathways (*Daneman & Engelhardt, 2017*; *Liebner et al., 2018*). Persistent peripheral inflammation in Parkinson's disease (PD) promotes BBB dysfunction, facilitating lymphocyte trafficking into the CNS-a phenomenon documented in PD patients. This BBB breakdown, combined with escalating neuroinflammation, establishes a reciprocal interaction between CNS and peripheral immune systems, characterized by increased leukocyte infiltration into brain parenchyma. During inflammatory states, unrestricted infiltration of peripheral immune cells through a compromised BBB orchestrates neuroinflammatory responses and neurodegenerative progression *via* combined paracrine and endocrine signaling mechanisms (*Sweeney, Sagare & Zlokovic, 2018*). Long regarded as an immunologically privileged compartment due to the BBB's separation from systemic circulation, the CNS maintains neuronal homeostasis while limiting immune cell trafficking. Nevertheless, recent insights into PD pathogenesis challenge this conventional view by demonstrating BBB dysfunction marked by endothelial cell derangement, basement membrane thickening, and vacuolar formation. Disrupted BBB integrity facilitates aberrant immune cell infiltration into CNS parenchyma, thereby propagating neuroinflammatory cascades and accelerating neurodegenerative processes.

## PD and α-Syn specific T cells

α-Syn assumes a central role in PD pathogenesis, serving as the major component of Lewy bodies-the pathological hallmark observed in postmortem PD tissues-while

driving neuroinflammation and neurodegeneration. This protein exists in two distinct conformational states: native monomers and helical membrane-bound conformers. Both conformers are prone to undergo conformational conversion into β-sheet-rich assemblies, stabilized *via* β-sheet interactions, which progress to form insoluble high-molecular-weight fibrils. These fibrillogenic intermediates ultimately polymerize into amyloid structures akin to those found in Lewy bodies (*La Vitola et al., 2021*). Endogenous α-Syn undergoes misfolding to form amyloid fibrils, which act as nucleation seeds promoting templated conversion of native α-Syn into pathological aggregates. These misfolded fibrils utilize axonal transport mechanisms to propagate across neuronal networks. Peripherally derived misfolded α-Syn fibrils are internalized by neurons, where they induce endogenous α-Syn aggregation *via* a templated seeding mechanism, exhibiting prion-like spreading properties through trans-synaptic transmission (*Holec & Woerman, 2021*). Increasing evidence indicates that Lewy body pathology is not limited to the CNS, involving extra-CNS compartments including sympathetic ganglia, cardiac plexus, and enteric neurons within the peripheral nervous system (PNS) (*Benarroch, 2007*; *Tanei et al., 2021*). Pathogenic α-Synuclein aggregates first accumulate in peripheral tissues, ascending through spinal projections to the midbrain before subsequently spreading to higher-order cortical regions in a pattern consistent with Braak staging criteria (*Braak et al., 2003*). PD pathophysiology is characterized by three cardinal immunological hallmarks: autoreactive T cell infiltration, dysregulated autoantigen presentation, and persistent microglial activation, documented in both clinical cohorts and preclinical models (*Sampson et al., 2016*; *Garretti et al., 2019*).

In α-Syn overexpressing transgenic murine models, perturbations in gut microbiota composition correlate with disease severity modulation, as germ-free animals exhibit delayed motor symptom onset compared to microbiota-intact controls. Notably, fecal microbiota transplantation from PD patients into germ-free mice recapitulates disease-specific motor deficits, providing direct evidence for microbiota-mediated pathogenesis (*Sampson et al., 2016*). The pathogenesis of PD has complex interactions with gut microbiota, abnormal aggregation of α-Syn, and neuroinflammation. The three form a vicious cycle through the gut-brain axis and jointly promote the progression of the disease. The composition of gut microbiota in patients with PD is significantly altered, with a decrease in the abundance of short-chain fatty acid (SCFA)-producing bacteria (such as Roseburia and Ruminococcus) and an increase in Gram-negative bacteria that produce lipopolysaccharide (LPS) (such as Enterobacteriaceae) (*Kalyanaraman, Cheng & Hardy, 2024*; *Mahbub et al., 2024*; *Yang et al., 2024*). This imbalance affects α-Syn through multiple pathways: on one hand, gut microbiota metabolites such as LPS and rhamnolipids can penetrate the intestinal barrier and enter the circulation, directly acting on neurons; on the other hand, the lack of SCFA weakens the expression of tight junction proteins (such as ZO-1 and occludin) in the intestinal and blood–brain barriers, facilitating the translocation of α-Syn and bacterial toxic products (such as LPS) into the brain and triggering neuroinflammation (*Ioghen et al., 2024*).

The aggregation of α-Syn activates microglia, causing them to polarize from anti-inflammatory M2 to pro-inflammatory M1 type, releasing cytokines such as TNF-α and IL-6, further damaging dopaminergic neurons. Moreover, SCFA produced by gut microbiota

can regulate the phenotype of microglia through histone deacetylase (HDAC) inhibition. Additionally, gut microbiota dysbiosis enhances the neurotoxicity of α-Syn by inducing oxidative stress (such as ROS/RNS generation) and ferroptosis, forming a "microbiota-inflammation-protein toxicity" positive feedback loop. In terms of therapeutic significance and intervention strategies, supplementing SCFA-producing bacteria or exogenous butyrate can improve intestinal barrier function, inhibit microglial activation, and reduce α-Syn aggregation by activating G protein-coupled receptors and the nuclear factor erythroid 2-related factor 2 (Nrf2) antioxidant pathway (*Sampson et al., 2016*; *Kalyanaraman, Cheng & Hardy, 2024*; *Nishiwaki et al., 2024*). Probiotic intervention has shown improvement in motor function and reduction in dopaminergic neuron loss in the substantia nigra in MPTP-induced PD mouse models. Inhibiting the Toll-like receptor 4 (TLR4)/NF-κB pathway can block LPS-induced neuroinflammation and reduce the pathological spread of α-Syn. Histone deacetylase inhibitors (such as butyrate) enhance the expression of antioxidant genes through epigenetic regulation and alleviate oxidative damage. The levels of SCFA in feces and the abundance of LPS-producing bacteria are positively correlated with PD clinical symptoms and can be used as early diagnostic indicators. In summary, gut microbiota, through the interaction of metabolic products and neuroimmune pathways, have become a key environmental factor in the pathogenesis of PD (*Zhou et al., 2024*; *Qi et al., 2025*; *Qu et al., 2025*; *Sharma et al., 2025a*). Targeted interventions on the microbiota-gut-brain axis (such as probiotics and SCFA supplements) provide new directions for the early prevention and treatment of PD, but further validation of the long-term safety and individual differences of microbiota regulation is needed.

The Braak hypothesis proposes that α-synucleinopathy and PD pathogenesis originate in the gastrointestinal tract and PNS, ascending to the CNS *via* the vagus nerve and dorsal motor nucleus of the vagus, thereby establishing a bidirectional pathological axis between enteric and central neuronal circuits. Emerging evidence demonstrates vagus nerve-associated peripheral denervation in PD (*Borghammer & Van DenBerge, 2019*), a pathological process that modulates immune responses *via* the cholinergic anti-inflammatory pathway, thereby influencing neuroimmune crosstalk critical for disease progression. This bidirectional signaling mechanism involves key components such as splenic macrophages, intestinal lymphocytes, and CNS nuclei, forming a neuroimmune axis that modulates inflammatory responses. In transgenic rodent models overexpressing α-Syn, pathological aggregates have been observed to propagate bidirectionally between enteric and CNS, further supporting the gut-brain axis hypothesis (*Ulusoy et al., 2017*). Motivated by PNS degeneration in PD, investigations into the gut-brain axis and microbiota's role in neurodegeneration have garnered significant attention. Notably, in rodent models utilizing rAAV-mediated α-Syn overexpression, midbrain substantia nigra-targeted α-Synucleinopathy induced enteric nervous system perturbations and microbiota compositional shift (*O'Donovan et al., 2020*). Braak's hypothesis proposes that pathologicalα-Syn initially originates in the gastrointestinal tract, with subsequent propagation to the CNS through vagal pathways. Immunohistochemical analyses detected α-Syn-positive staining within colonic neural plexuses exclusively in treatment-naïve individuals in the initial phases of PD, while such pathological markers were absent in

both asymptomatic controls and irritable bowel syndrome (IBS) cohorts (*Shannon et al., 2012*). After 1.5 months of intragastric low-dose rotenone exposure in mice, α-Syn accumulation was observed in the peripheral nervous system, dorsal motor nucleus, and medial lateral spinal cord nucleus without observable motor dysfunction. Following an additional 2 months of exposure, intestinal dysregulation emerged as a cardinal phenotype. By 3 months, α-Syn aggregation and dopaminergic neuronal loss became evident in the midbrain substantia nigra, culminating in the manifestation of PD-like motor deficits 20098733 (*Pan-Montojo et al., 2010*).

Additionally, within this model, α-Syn secreted from gut neurons could be internalized by presynaptic sympathetic nerve cells and retrogradely transported to neuronal populations (*Pan-Montojo et al., 2012*). Rotenone-induced PD models *via* intragastric administration have been shown to faithfully recapitulate the spatio-temporal progression of both neuropathology and clinical manifestations, corroborating the Braak Hypothesis postulating peripheral initiation of α-Syn pathology with subsequent retrograde propagation through the CNS. PD patients exhibited upregulated concentrations of pro-inflammatory cytokines including TNF-α, IL-1β, Interleukin-6 (IL-6), and interferon-γ (IFN-γ), which demonstrated inverse correlations with clinical duration of illness (*Devos et al., 2013*). Notably, colonic mucosal CD4+ T cell infiltration density demonstrated significant elevation in constipated PD patients compared to non-constipated counterparts (*Chen et al., 2015*). The gastrointestinal tract potentially serves as the primary locus for inflammatory and pathological initiation, while simultaneously representing the anatomical region where adaptive immune responses are activated to counter α-Syn aggregate accumulation.

Notably, emerging evidence demonstrates that T cell reactivity toward α-Syn in PD patients precedes clinical diagnosis, exhibiting a biphasic response characterized by peak immunoreactivity during prodromal phases followed by progressive attenuation with disease advancement, thereby establishing stage-dependent modulation of adaptive immunity (*Bryant et al., 2015*; *Lindestam Arlehamn et al., 2020*). Peripheral APCs may initiate detection of progressive α-Syn elevations, with chronic exposure culminating in neurotoxic consequences. Experimental models further demonstrate α-Syn exerts chemotactic activity capable of recruiting neutrophils and monocytes toward pathological foci (*Stolzenberg et al., 2017*). During neuroinflammatory episodes, peripherally primed APCs (*e.g.*, monocytes) exhibit tissue-transcending migratory capacity, including cerebral transmigration (*Ransohoff, 2011*). PD pathophysiology involves microglial activation triggering sequential initiation of the classical complement cascade, targeted clearance of compromised neurons, and subsequent BBB disruption that promotes B lymphocytes and T lymphocytes transmigration into lesioned areas, culminating in CNS-directed autoimmunity. Mechanistically, reactive microglia facilitate α-Syn post-translational modifications through nitrosative/oxidative pathways, ultimately driving selective degeneration of dopaminergic neuronal populations (*Shavali, Combs & Ebadi, 2006*). Circulating α-Syn monomers evade immune surveillance in peripheral circulation, yet their structural transition into stable dimeric/oligomeric conformations triggers pathogen-associated molecular pattern (PAMP) recognition, initiating phagocyte recruitment

and neuroinflammatory cascade. These pathological aggregates compromise neuronal homeostatic equilibrium and synaptic integrity, while functioning as endogenous chemoattractant signals upon liberation from necrotic dopaminergic neurons into extracellular compartments, thereby orchestrating targeted microglial migration to neurodegeneration epicenters (*Lingor, Carboni & Koch, 2017*). Sustained impairment in clearing neurotoxic α-Syn aggregates compromises proteostasis, driving chronic neuroinflammation through microglial MHC II hyperactivation and maladaptive antigen presentation, ultimately culminating in progressive neuronal degeneration (*Wang, Liu & Zhou, 2015*).

Under physiological conditions, α-Syn intrinsically regulates synaptic vesicle cycling, maintains mitochondrial bioenergetics, and coordinates cellular stress adaptation. Pathologically, post-translational modifications-particularly oxidative modifications, phosphorylation (*e.g.*, pS129), and tyrosine nitration (notably Y39)-fundamentally alter α-Syn conformation, accelerating protofibril formation and Lewy pathology maturation (*Barrett & Timothy Greenamyre, 2015*). Structural alterations in α-Syn conformation expose cryptic epitopes that prime peripheral T cell-mediated adaptive immunity, mechanistically linking autoimmune activation to nigrostriatal pathway degeneration (*Benner et al., 2008*). The protein intrinsically activates both microglia and APCs, establishing bidirectional neuroimmune communication between CNS and peripheral compartments. Crucially, α-Syn-reactive T lymphocytes demonstrate cerebrovascular transmigration capacity, undergoing antigen-driven reactivation upon encountering MHC-presenting microglial populations within the substantia nigra. Experimental α-Syn overexpression models-whether achieved through viral vector-mediated delivery or transgenic manipulation-consistently exhibit early-stage neuroinflammation characterized by microglial proliferation and lymphocyte infiltration into midbrain regions, preceding detectable dopaminergic neuron loss (*Sanchez-Guajardo et al., 2013*). These findings position α-Syn as a central etiological driver in PD pathogenesis while providing mechanistic rationale for immunotherapeutic strategies targeting α-Syn clearance, thereby establishing a novel therapeutic paradigm for neurodegenerative disease intervention.

T lymphocytes undergo antigen-driven activation *via* cognate interaction with microglial MHC class II molecules complexed with epitope-bearing antigens, a process mediated through Human Leukocyte Antigen (HLA) polymorphic loci that dictate the repertoire of presented antigens. Allelic variations within these HLA genes determine preferential presentation of specific pathogenic epitopes, including those derived from α-Syn aggregates. α-Syn-derived epitopes demonstrate enhanced immunogenicity, preferentially driving antigen-specific T cell activation through high-affinity T cell receptor (TCR) engagement. The HLA system, encoding the polymorphic components of MHC molecules, constitutes the most genetically diverse region in humans, harboring >100 immune-related loci that orchestrate antigen presentation and confer susceptibility to both autoimmune disorders and neurodegenerative conditions including PD. The HLA system orchestrates adaptive immunity through its encoded MHC class I/II molecules, specialized in presenting proteolytic peptides to cognate T cell receptors. GWAS revealed 17 risk loci demonstrating significant pleiotropy between PD and autoimmune disorders,

mechanistically linking aberrant immune activation to neurodegenerative cascades in PD pathogenesis. Genetic susceptibility loci at HLA-DRB5 and HLA-DQB1 underpin convergent immunopathological mechanisms linking PD with autoimmune disorders including Crohn's disease, ulcerative colitis, and rheumatoid arthritis. Functionally, HLA-DQB1 encodes β-chains critical for extracellular antigen presentation *via* MHC class II complexes. This genetic pleiotropy defines a pathogenic continuum where chronic immune activation drives selective degeneration of dopaminergic neurons within the nigrostriatal pathway. GWAS have conclusively established HLA allelic diversity as a major modifiable risk factor for PD, with specific non-coding variants such as rs3129882 in HLA-DRA acting as regulatory elements that modulate neuroinflammatory responses through MHC class II-mediated antigen presentation dynamics (*Hamza et al., 2010*; *Wissemann et al., 2013*).

*Sulzer et al. (2017)*'s seminal work identified clonally expanded α-Syn-reactive T cell pools in PD patients, demonstrating selective epitope targeting encompassing the Y39 phosphorylation site and S129 ubiquitination domain. Mechanistically, APC process and present Y39-flanking polypeptides, triggering coordinated cytokine production by both cytotoxic CD8+ T lymphocytes and CD4+ Th subsets. Structural analyses revealed preferential binding of Y39-derived epitopes to HLA-DRB1*1501 and HLA-DRB5*01:01 molecules, two Class II variants with established autoimmune susceptibility profiles. Such pathogenic processing of endogenous α-Syn generates cryptic epitopes that evade thymic selection, enabling cognate HLA presentation to clonally escaped T cell clones. This immune tolerance breach mechanism shares striking parallels with prototypical autoimmune disorders like type 1 diabetes, where similar HLA-epitope-T cell triad interactions drive pathology. While neurodegenerative conditions traditionally lack canonical autoimmune classification, the observed thymic selection failure coupled with pathogenic protein processing suggests a hybrid mechanism bridging misfolding pathology and immune dysregulation. Experimental validation revealed that substantia nigra dopamine neurons upregulate MHC I expression *via* cytokine signaling cascades triggered by α-Syn/neuromelanin-activated microglia, rendering them susceptible to CD8+ T cell-mediated cytotoxicity through peptide-MHC I complex recognition. GWAS have identified more than 20 genetic loci associated with familial PD pathogenesis (*Hernandez, Reed & Singleton, 2016*), with functional enrichment analyses showing predominant involvement in lysosomal clearance mechanisms. Pathogenic variants in α-Syn (particularly dopamine-modified isoforms) and LRRK2 dysregulate chaperone-mediated autophagy—an age-compromised proteostasis pathway critical for α-Syn turnover. Notably, extracellular α-Syn oligomers exhibit prion-like propagation capacity, being internalized *via* endocytic pathways during PD progression to nucleate pathological aggregation cascades in recipient neural cells (*Cuervo et al., 2004*; *Luk et al., 2012*).

To elucidate the pathophysiological connection between PD pathogenesis and T-cell reactivity, longitudinal assessments of α-synuclein-specific T-cell dynamics were performed in relation to motor symptom onset within clinical cohorts. Analyses revealed progressive enhancement of α-Syn-directed T cell responses in PD patients, peaking near the proximity to clinical diagnosis. Furthermore, these reactive T cell profiles correlated with multidimensional immune signatures encompassing proinflammatory cytokines,

immunomodulatory factors, and secretory cell markers. Importantly, age-stratified analyses revealed an age-dependent augmentation of α-synuclein-specific T-cell responses in PD patients, in contrast to stable reactivity profiles observed in age-matched healthy individuals. Sex-stratified evaluations further demonstrated enhanced α-Syn-directed T-cell reactivity in male PD subjects, consistent with the established male predominance in PD prevalence (*Arvey et al., 2020*; *Williams et al., 2024*). Longitudinal profiling demonstrated a temporal decline in α-Syn-directed immune responses post-diagnosis, concurrent with an age-dependent amplification of reactivity (*Rauschenberger et al., 2022*; *Sirerol-Piquer et al., 2025*). This dissociation from chronological aging refutes nonspecific immunosenescence and instead supports age-associated autoimmune priming. No significant associations emerged between T cell reactivity and clinical metrics (MoCA/UPDRS scores), suggesting these immune responses predominantly occur during pre-symptomatic stages, akin to prototypical autoimmune mechanisms. The temporal coupling of α-Syn reactivity to both preclinical phases and diagnostic proximity reinforces its etiological role in PD pathogenesis. Critically, these reactive T cell signatures exhibit phase specificity, serving as biomarkers for preclinical and early motor-stage PD. This supports targeted monitoring of α-Syn-specific T cell dynamics in prodromal populations (*e.g.*, REM sleep behavior disorder cohorts) to stratify individuals for preemptive neuroimmunological interventions (*Lindestam Arlehamn et al., 2020*).

Emerging data implicate proteostasis failure—particularly involving α-Syn clearance—in generating cryptic epitopes that drive age-dependent immune activation in PD pathogenesis. A paradigm shift is occurring in PD research, with growing recognition of neuroimmune axis dysregulation as central to disease progression, positioning immunomodulation as a strategic therapeutic frontier. Chronotherapeutic precision is paramount: Neuroprotective interventions must precede the neurotoxic cascade triggered by peripheral inflammation, α-Syn oligomerization, blood–brain barrier compromise, and cytotoxic T cell infiltration. Therapeutic windows of opportunity exist during early tolerance breakdown to α-Syn epitopes, prior to full-blown autoimmune targeting of dopaminergic neurons. Implementing preclinical diagnostic platforms to detect α-Syn-reactive T cell clones and quantify neuroinflammatory trajectories could enable stage-specific immunotherapy administration, effectively arresting PD progression before irreversible neurodegeneration occurs.

## Application of Tregs in PD therapy

Tregs play an essential role in maintaining systemic immune tolerance. These thymus-derived lymphocytes home to peripheral tissues and mechanistically inhibit autoreactive T cell activation and expansion through three core pathways: (1) secretory activity of immunosuppressive cytokines such as Transforming growth factor beta (TGF-β), Interleukin-10 (IL-10), and Interleukin-35 (IL-35); (2) granzyme B/perforin-mediated induction of cytotoxicity and apoptotic cell death; and (3) surface-displayed immunoregulatory molecules, notably Cytotoxic T lymphocyte-associated antigen 4 (CTLA4), CD39, and CD73. By disrupting effector T cells (Teffs) metabolism and impairing APC maturation, Tregs maintain immune equilibrium and prevent

pathological autoimmunity (*Yao et al., 2021*). Tregs exhibit significant functional links to neurodegenerative pathophysiology, as evidenced in PD models (*Qian et al., 2008*). These lymphocytes confer neuroprotection by suppressing microglial hyperactivation toward misfolded and post-translationally modified α-Syn aggregates. Conversely, Tregs insufficiency may amplify α-Syn-directed autoimmune responses through failed tolerance mechanisms. PD patients exhibit significantly diminished circulating levels of IL-10 and TGF-β relative to healthy controls, whereas serum IL-17 concentrations display a pathologically elevated profile. This cytokine imbalance underscores a Th17-polarized inflammatory state, indicative of compromised immunoregulatory feedback in PD pathogenesis (*Han et al., 2015*). Functional assessments revealed compromised Treg cell-mediated suppression of Teffs activity in PD patients, contrasted with preserved proliferative responses in both naive T cells and Teffs populations. These immunophenotypic alterations, when contextualized with emerging evidence of gut-barrier dysfunction-including elevated intestinal permeability, colocalization of phosphorylated α-Syn/Lewy pathology, and persistent enteric inflammation in PD-converge to substantiate the dual-hit pathogenesis. This mechanistic framework posits synergistic peripheral-CNS interactions, where peripherally derived α-Syn antigens and microbial-inflammatory triggers potentiate neuroimmune cascades driving PD progression (*Saunders et al., 2012*).

TGF-β demonstrates neuroprotective capacity by preserving nigrostriatal dopaminergic neurons, while Tregs antagonize Th17-mediated neurotoxicity in the substantia nigra. PD patients exhibit a marked elevation in circulating Th17 populations, concurrently exhibiting compromised Tregs functionality-evidenced by reduced Foxp3 transcription and impaired immunosuppressive control over Teffs cell activation. This Th17/Treg disequilibrium correlates with progressive neurodegeneration, suggesting a pathogenic axis where diminished regulatory capacity permits unchecked neuroinflammatory cascades (*Reynolds et al., 2010*). The Treg/Th17 imbalance disrupts neuroimmune homeostasis, accelerating PD pathogenesis through dual-phase mechanisms. During prodromal stages, BBB compromise permits Tregs infiltration that transiently modulates neuroinflammation. Progressive erosion of Treg-mediated immunosuppression drives a pathogenic polarization toward IFN-γ-secreting Th1 and IL-17-producing Th17 dominance, disrupting the Tregs/Teffs equilibrium and precipitating systemic immune tolerance failure in advancing PD (*Saunders et al., 2012*). Beyond suppressing Teffs proliferation and effector functions, Tregs orchestrate glial cell homeostasis through IL-4, IL-10, and TGF-β signaling, enforcing anti-inflammatory microglial states and neuroprotective astrocyte polarization (*Ito et al., 2019*). This regulatory axis simultaneously suppresses microglial ROS generation while augmenting astrocyte-derived neurotrophic mediator release, notably brain-derived neurotrophic factor (BDNF), thereby promoting neuronal survival (*Benner et al., 2004*). Given the pathogenic Tregs/Teffs imbalance in PD progression, therapeutic strategies augmenting Tregs populations *via* positive regulators have demonstrated neuroprotection in preclinical models, though their molecular mechanisms remain unresolved.

In MPTP-induced PD mice, bee venom therapy selectively expanded CD4+CD25+Foxp3+ Tregs populations, correlating with attenuated dopaminergic degeneration (*Chung et al., 2012*). Similarly, bacille Calmette-Guerin (BCG) vaccination elicited Foxp3+

Tregs expansion in the same model, mediating neuroprotective effects through immunomodulation. Both interventions highlight the therapeutic potential of Tregs amplification, despite incomplete understanding of downstream signaling pathways governing these protective outcomes (*Lacan et al., 2013*). Granulocyte-macrophage colony-stimulating factor (GM-CSF) enhances CD4+CD25+Foxp3+ Tregs expansion, conferring neuroprotection against MPTP-induced dopaminergic toxicity. Adoptive transfer of Copolymer 1-primed splenocytes and lymph node-derived T cells into MPTP-model mice recapitulates this neuroprotective phenotype through Treg-dependent mechanisms. Pharmacological activation of vasoactive intestinal peptide receptor 2 (VIPR2) amplifies GM-CSF production, driving Th2 anti-inflammatory polarization and Tregs proliferation, thereby attenuating neurodegeneration in PD models (*Kosloski et al., 2013*).

## CONCLUSION

α-synuclein's pathological role in Parkinson's disease makes it a prime target for disease-modifying therapies. Emerging immunotherapies-including monoclonal antibodies and epitope-specific vaccines-aim to block α-Syn's neurotoxic aggregation and spread. By reducing α-Syn-driven neuroinflammation, these approaches seek to preserve dopaminergic neurons and ultimately slow PD progression (*Nasrolahi et al., 2019*; *Weihofen et al., 2019*). This study supports α-Syn-targeting therapies for PD. However, antibody-based treatments face challenges including poor BBB penetration, inflammation, and off-target effects. Immunopathology is central to PD progression, with preclinical evidence showing BBB disruption, gut α-Syn accumulation, and neuroinflammation preceding motor symptoms. These features coincide with microglial activation and immune involvement in the nigrostriatal system. As PD's defining hallmark, α-Syn triggers neuroinflammation and autoimmunity *via* T-cell activation. Current therapies mainly address symptoms, while immunomodulatory strategies may slow disease progression. Preclinical data indicate boosting Tregs reduces neurodegeneration, though their precise mechanisms and potential side effects require further investigation.

The Teffs/Tregs cell imbalance in PD pathogenesis requires further study. Peripheral immune infiltration and microglial activation drive chronic neuroinflammation, accelerating dopaminergic neuron loss. PD patients exhibit α-Syn-reactive T cells targeting abnormally modified α-Syn epitopes that evade immune tolerance. Within PD-affected brain regions, α-Syn aggregates recruit these autoreactive T cells alongside activated microglia and antigen-presenting cells. Critically, these T cells colocalize with α-synucleinopathy and fibril formation, supporting PD's autoimmune etiology. This reveals therapeutic opportunities for T cell-targeted approaches like checkpoint inhibitors or tolerance induction. Engineered α-Syn-specific Tregs represent a novel strategy to suppress autoimmunity both peripherally and within the substantia nigra. Understanding α-Syn-reactive T cell dynamics, BBB integrity, and regulatory networks could yield stage-specific interventions. Given current therapies' limitations, Treg-based immunomodulation offers transformative potential to delay PD onset and modify disease progression. Our present work systematically examines the pathophysiological interplay

involving neuroinflammatory cascades, BBB integrity loss, CNS immune regulation, and T cell subpopulations (including Tregs) during PD progression, establishing an evidence-based framework for advancing immunotherapeutic strategies. These mechanistic insights highlight the transformative potential of cellular immunity in developing novel therapeutic paradigms for PD management.

## Abbreviations

| | |
|---|---|
| PD | Parkinson's disease |
| α-Syn | α-Synuclein |
| CNS | Central nervous system |
| MS | Multiple Sclerosis |
| AD | Alzheimer's disease |
| Tregs | Regulatory T cells |
| PET | Positron Emission Computed Tomography |
| RBD | Rapid eye movement sleep behavior disorder |
| MHC II | Major histocompatibility complex II |
| GWAS | Genome wide association study |
| LRRK2 | Leucine-rich repeat kinase 2 |
| TNF-α | Tumor necrosis factor-α |
| CCL5 | C-C motif chemokine ligand 5 |
| Th | Helper T cells |
| NF-κB | Nuclear factor kappa-B |
| SNpc | Substantia nigra pars compacta |
| NK cells | Natural killer cells |
| CSF | Cerebrospinal fluid |
| MPTP | 1-Methyl-4-phenyl-1,2,3,6-tetrahydropyridine |
| iPSCs | Induced Pluripotent Stem Cells |
| MitAP | MITRE Text and Audio Processing |
| MHC I | Major histocompatibility complex I |
| DC | Dendritic cells |
| APC | Antigen-presenting cells |
| BBB | Blood brain barrier |
| HLA | Human Leukocyte Antigen |
| MoCA | Correlations with cognitive function |
| CTLA4 | Cytotoxic T lymphocyte-associated antigen 4 |
| TGF-β | Transforming growth factor beta |
| IL-10 | Interleukin-10 |
| IL-35 | Interleukin-35 |
| IL-2 | Interleukin-2 |
| IL-6 | Interleukin-6 |
| PINK1 | PTEN induced putative kinase 1 |
| HDAC | Histone deacetylase |
| Teffs | Effector T cells |
| TLR4 | Toll-like receptor 4 |
| GM-CSF | Granulocyte-macrophage colony-stimulating factor |

| SCFA | Short-chain fatty acid |
|---|---|
| IBS | Irritable bowel syndrome |
| IFN-γ | Interferon-γ |
| PAMP | Pathogen-associated molecular pattern |
| TCR | T cell receptor |
| AAV | Adeno-associated virus |
| Nrf2 | Nuclear factor erythroid 2-related factor 2 |
| BCG | Bacille Calmette-Guerin |
| VIPR2 | Vasoactive intestinal peptide receptor 2 |
| PNS | Peripheral nervous system |

### Funding

This study was supported by the National Nature Science Foundation of China (Grant No. 82304434), GuangDong Basic and Applied Basic Research Foundation (Grant No. 2023A1515111199), China Postdoctoral Science Foundation (Grant No. 2022M713263), Scientific Research Starting Foundation for High-level Talents of Meizhou People's Hospital (Grant No. KYQD202501), Shining Across China-Medicinal Research Fund (Grant No. Z04J2023E095), Social Development Science and Technology Plan Project of Meizhou (Grant No. 2024C0301079). The funders had no role in study design, data collection and analysis, decision to publish, or preparation of the manuscript.

### Grant Disclosures

The following grant information was disclosed by the authors:
National Nature Science Foundation of China: 82304434.
GuangDong Basic and Applied Basic Research Foundation: 2023A1515111199.
China Postdoctoral Science Foundation: 2022M713263.
Scientific Research Starting Foundation for High-level Talents of Meizhou People's Hospital: KYQD202501.
Shining Across China-Medicinal Research Fund: Z04J2023E095.
Social Development Science and Technology Plan Project of Meizhou: 2024C0301079.

### Competing Interests

The authors declare there are no competing interests.

### Author Contributions

- Zhuomiao Lin conceived and designed the experiments, performed the experiments, authored or reviewed drafts of the article, and approved the final draft.
- Xihui Yu performed the experiments, prepared figures and/or tables, and approved the final draft.
- Yunming Zhong performed the experiments, prepared figures and/or tables, and approved the final draft.

- Guozhu Tan analyzed the data, prepared figures and/or tables, and approved the final draft.
- Jiahong Zhong conceived and designed the experiments, analyzed the data, authored or reviewed drafts of the article, and approved the final draft.

## Data Availability

This is a literature review.

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
