# Peer review of "The role of T cells in the treatment of Parkinson’s disease"

_PeerJ, doi:10.7717/peerj.19818_

## Round 0.1 · original submission · Major Revisions

Reviewer 1 ·

Basic reporting

I find that the review deals with interesting topics and that it is a starting point for further reflections and insights.
In the "additional comments" section you can find some points that I think should be managed for a simpler and more immediate reading.

Experimental design

I think sub-paragraphs should be inserted to better divide the topics covered.

The conclusion should be formulated and shortened because they are excessively verbose.

Validity of the findings

The work is interesting as studies related to Parkinson's disease and acquired immunity are gaining increasing relevance.

Additional comments

Line 50: I would delete this sentence.
Lines 84-89: I would insert this sentence in the conclusion.
Line 108: insert a comma after word "presentation".
Line 128: I would separate the paragraph from the previous one and put as title "PD and microglia"
Lines 150-155: simplify this sentence.
Lines 155-167: for a better understanding, I think that these lines should be inserted after line 169. Furthermore, I think that the paragraph from line 169 (Activated glia secrete...) to 173 should be inserted as an introductory cap to the paragraph "PD and T cells".
Lines 175-176: I would delete this sentence because it is redundant.
Lines 185-187: I would delete this sentence because it doesn't seem useful in the context of the discussion.
Lines 222-228: I would delete this sentence because it doesn't seem useful in the context of the discussion.
Line 228: I would insert a new paragraph: "PD and cytokines".
Line 330: The topic synuclein-microbiota is very current and interesting, I would consider inserting a new separate paragraph.
Line 375: insert a new sub-paragraph.
Line 418: insert a new sub-paragraph.
Lines 464-487: insert before line 442 (after line 441).
Line 579: Insert a capital letter after the dot ("however").
Lines 606-633: choose whether to always start with a capital or lowercase letter and make all the wording uniform.

·

Basic reporting

Authors describe in a good Perspective the overall landscape of the neuroimmune interactions on Parkinson’s disease, with special emphasis on T-Cells, Microglia, and TH-dopaminergic neurons crosstalk. Saying that and having in consideration that this is a review paper, I feel that one of the weakest points is on the literature citation. There are citations missing for findings or statements described on this manuscript. Specially on the intro but extending into the rest of the text.

These are a list of Improvements I would Include to enhance the manuscript:

Line 48, citate the epidemiological studies

Line 69, citate the papers showing MS, AD and PD inflammatory activation

Line 73, there are different papers with contradictory results, authors should cite

Line 77, on “Syn-specific T cells” I would suggest citing: David Sulzer & Alessandro Sette 2020, Nature communications

Line 149, authors should describe MHC I upregulation on dopaminergic neurons

Line 167, citate lit. about upregulated MHC II on microglia

Line 183, cite the “Quantitative analysis”

Line 187, “T-cell infiltration exclusively observed in pathological specimens.” That is not 100%, neurological controls could have some degree of T-cells on the brain parenchyma, at some degree, cite papers

Line 196, cite the paper referent to the different ratios

Line 199, cite the study for postmortem neuropathological evaluations

Line 229, I should specify RANTES and eotaxin as CCL5 and CCL11 for more general public reading the review

Line 260, cite Aberrant LRRK2 study

Line 264, Authors first describe that microglia activation promotes T cell infiltration and now that is the T cell infiltration who triggers brain microglia activation, please clarify

Line 287, Perivascular microglia, or perivascular macrophages? Cite

Line 324, cite study

Line 330, “documented in both clinical cohorts dans preclinical models” cite the studies

Line 375, I would cite “α-Synuclein-specific T cell reactivity is associated with preclinical and early Parkinson’s disease” Nat coms 2020

Line 386, be consistent B and T, or B/T

Line 429, cite after the first statemen, also define GWAS acronym as it is used further on the manuscript

Line 450, cite sex-stratified paper

Line 452, cite longitudinal profiling paper

Line 462, use “RBD” since its already described

Line 512, authors some time uses “Tregs cells” and other time “Tregs”, be concordant

Line 519, for clarity, use either Tnaiv and Teff or full name

Line 547, what is BCG?

Line 595, rewrite, difficult to understand

Line 605, work on the LIST OF ABBREVIATIONS, missing abbreviation like PNS, PAMP,…

Line 655, REFERENCES, I don’t like the way authors repeat the same paper a, b, c, ... as they cite different parts of the review. I would add only one entry on the references.

Experimental design

The Study design is good and they follow the standards for paper review, I would suggest to include more newly published papers since this topic is on a high trend and there are many new publications.

Validity of the findings

No objections on the conclusions, maybe Integrate better the 2 Figures this paper have.

---

## Round 0.2 · Minor Revisions

As you will see, both peer reviewers were very positive about the paper following your changes. They are both happy for the article to be accepted following some very minor corrections.

Reviewer 1 ·

Basic reporting

Dear authors,

Thank you for accepting my suggestions and actively modifying your article.

I have carefully reviewed everything, and I am satisfied.

I only ask you to double-check the numerical sequence that appears on line 509 to make sure it is not an error.

Experimental design

Same comments as the previous draft

Validity of the findings

Same comments as the previous draft

·

Basic reporting

The authors have addressed all comments professionally and thoroughly.

I recommend the article for acceptance; however, minor proofreading is still needed, for example, consistency in terms such as “T cells” vs. “T-cells.”

Experimental design

-

Validity of the findings

-

---

## Round 0.3 · accepted · Accept

Thank you for submitting the final corrections. This article is now ready for publication.